# Peer review of "Neuropsychological Insights into Coping Strategies: Integrating Theory and Practice in Clinical and Therapeutic Contexts"

_2673-5318, doi:10.3390/psychiatryint5010005_

Round 1
Reviewer 1 Report
Comments and Suggestions for Authors
Thank you for the comprehensive review to bridge the gap in coping strategies between theory and practice. I appreciate the well-thought organization of the results. Here are some suggestions and considerations:
Abstract
Lines 8-9 sentence: consider rewriting for clarity
Lines 11, 13, 17: hyphenated words (impact; interact): remove hyphen (this happens throughout the paper and distracts from your poignant findings)
Materials and Methods
Be more explicit as to the scope of this narrative review (as opposed to an exhaustive systematic literature review). I do appreciate the details regarding the databases and inclusion criteria. Was there a reporting standard (e.g., PRISMA or other) that was followed? Would be interesting to also note the initial results and how you arrived at the results
Results
Would recommend a diagram or model to illustrate the pathways and relationships between coping mechanisms; and/or ways to assess and measure these mechanisms
Line 159: COPE scale- is it an acronym? Write out what it means and be consistent (lower case in Line 165 for example).
Great organization of findings.
Discussion
Line 623: remove “&” and replace with “and”
The section on your limitations was well-appreciated; would recommend highlighting once more towards the end why this works was needed and that the scope was to not provide an exhaustive review.
Comments on the Quality of English LanguageSee my recommendations above (Re- hyphenated words).
Author Response
Dear Reviewer,
thank you very much for your valuable comments.
Enclosed is our response. Please note that 5 figures are included in the revised manuscript.
Best regards
Dr Maria Theodoratou & Prof. Marios Argyrides

Reviewer 2 Report
Comments and Suggestions for Authors
The manuscript by Theodoratou and Argyrides reviews the state of the art of copying strategies. Nevertheless, the work exhibits significant flaws that require correction before publication.
In general, the writing should be more concise and less circumlocutory. It is advisable to synthesize and go straight to the data for better understanding so that the reader does not get lost.
The results should be presented in a more systematic manner, distinct from being embedded within the text. This approach makes it challenging to extract the results. To align with the integrative nature stated in the title, novel findings or conclusions should be derived by effectively integrating various studies from the review.
In the review, it is necessary to include at least one graph or diagram, and at least one table that includes the systematically reviewed works and their characteristics, as it is stated that they have been reviewed systematically.
Comments on the Quality of English LanguagePlease rectify typos (for example, 'ap-plication' in the abstract section) and refine English grammar and style with the expertise of a native speaker.
Author Response
Thank you for your thorough review and constructive criticism of our manuscript. We recognize the importance of your comments and agree that it is essential to improve the clarity and structure of our paper. Accordingly, we have made major revisions that are included in the attached file and the revised manuscript. We have also corrected the typos.
We believe that these changes address your concerns and significantly improve the manuscript, making the review more accessible and informative to the reader. Thank you again for your comments to improve the review.
Sincerely,
Dr Maria Theodoratou -Prof. M. Argyrides

Reviewer 3 Report
Comments and Suggestions for Authors
In this review, the authors have discussed the application and significance of coping strategies within the domains of clinical psychology and neuropsychology. The mentioned six cognitive coping mechanisms high-light the points of this review.
No comments from me.
Sincerely
Author Response
Dear Reviewer,
Thank you for your comments on our review of the use and importance of coping strategies in clinical psychology and neuropsychology. We are pleased to hear that you found the discussion of the six cognitive coping mechanisms to be a highlight and central to the points of our review.
Your confirmation that no other comments or concerns were raised is greatly appreciated. It reassures us that our efforts to address this topic comprehensively have been successful. Should you have any suggestions or insights in the future, we would be happy to consider them to further improve our work.
Thank you again for your time and valuable feedback.
Best regards,
Dr Maria Theodoratou & Prof. Marios Argyrides
Reviewer 4 Report
Comments and Suggestions for Authors
Dear Editor,
I appreciate the opportunity to review the manuscript entitled:
"Neuropsychological Insights into Coping Strategies: Integrating Theory and Practice in Clinical and Therapeutic Contexts"
I commend the authors for describing this critical and timely issue. The paper is engaging and well-written; however, I would like to highlight some issues that merit revision:
Page 6 of the manuscript notes.: "The prefrontal cortex is essential for complex cognitive functions and for moderating stress responses through its interactions with limbic structures. This neural nexus is critical in mediating immediate stress responses and maintaining the balance between different bodily systems. This is correct, but the manuscript does not detect whether an assessment of what could be an additional factor in the therapeutic diagnostic process, i.e., neuroimaging, has been conducted. Increasingly, modern neuroimaging techniques are being used in complex cases both in diagnosis and in therapeutic follow-up. I would ask the authors if this aspect was evaluated during the investigation and add a short paragraph on this issue; if data are unavailable, add them to the limitations.
Author Response
Dear Reviewer,
Thank you for your appreciation of our manuscript and for recognizing the importance and timeliness of the topic we have discussed. We are pleased to know that you found the paper engaging and well written.
We also appreciate your insightful feedback regarding the inclusion of neuroimaging as a potential factor in the therapeutic diagnostic process. Indeed, your suggestion about the increasing relevance of modern neuroimaging techniques in both diagnosis and therapeutic follow-up in complex cases is an important one.
On page 6, while we discuss the role of the prefrontal cortex and its interactions with limbic structures, we did not explicitly address the neuroimaging aspect. This omission, as you rightly point out, may be an area for further exploration.
In response to your suggestion, we have found data to determine whether neuroimaging aspects were evaluated. So, we are including a paragraph discussing how neuroimaging techniques were used and their impact on our findings.
Thank you again for your constructive feedback. It is immensely valuable to us in improving the quality and scope of our work.
(PLEASE NOTE AFTER REWRITING SOME PARAGRAPHS IN MORE CONDENSED STYLE, page 6 is now page 4).

Round 2
Reviewer 2 Report
Comments and Suggestions for Authors
The manuscript version has been improved from the previous one and is now ready for publication. Please ensure that the figures are in the correct format, and all images and templates used are freely available for use (CC license or similar) to avoid any legal issues.